# Research Progress on the Effect and Mechanism of Tea Products with Different Fermentation Degrees in Regulating Type 2 Diabetes Mellitus

**DOI:** 10.3390/foods13020221

**Published:** 2024-01-10

**Authors:** Guangneng Li, Jianyong Zhang, Hongchun Cui, Zhihui Feng, Ying Gao, Yuwan Wang, Jianxin Chen, Yongquan Xu, Debao Niu, Junfeng Yin

**Affiliations:** 1College of Light Industry and Food Engineering, Guangxi University, Nanning 530003, China; 2Tea Research Institute, Chinese Academy of Agricultural Sciences, Hangzhou 310008, China; zjy5128@tricaas.com (J.Z.);; 3Tea Research Institute, Hangzhou Academy of Agricultural Sciences, Hangzhou 310024, China

**Keywords:** type 2 diabetes, different fermentation degree of tea, tea products, biological activity, blood glucose

## Abstract

A popular non-alcoholic beverage worldwide, tea can regulate blood glucose levels, lipid levels, and blood pressure, and may even prevent type 2 diabetes mellitus (T2DM). Different tea fermentation levels impact these effects. Tea products with different fermentation degrees containing different functional ingredients can lower post-meal blood glucose levels and may prevent T2DM. There are seven critical factors that shed light on how teas with different fermentation levels affect blood glucose regulation in humans. These factors include the inhibition of digestive enzymes, enhancement of cellular glucose uptake, suppression of gluconeogenesis-related enzymes, reduction in the formation of advanced glycation end products (AGEs), inhibition of dipeptidyl peptidase-4 (DPP-4) activity, modulation of gut flora, and the alleviation of inflammation associated with oxidative stress. Fermented teas can be used to lower post-meal blood glucose levels and can help consumers make more informed tea selections.

## 1. Introduction

Tea is one of the most popular beverages in the world. According to different processing methods, tea can be divided into six categories: green tea (non-fermented tea), white tea and yellow tea (slightly fermented tea), oolong tea (moderately fermented tea), black tea (fully fermented tea), and dark tea (post-fermented tea) [1]. The primary basis for these classifications lies in the extent of fermentation, which plays a fundamental role in comprehending the vast realm of tea. Tea has been extensively and comprehensively investigated, particularly regarding its impact on the regulation of glucose and lipid metabolism. In recent years, the field of tea and health research has placed significant emphasis on the prominent aspect of tea’s health benefits. Similarly, Kombucha tea deserves attention, particularly in blood glucose regulation. It is produced through the co-cultivation of yeast, lactic acid bacteria, and acetic acid bacteria with tea of varying fermentation levels as the raw material [2,3,4]. Due to its rich content of tea polyphenols and organic acids [3], it can effectively reduce fasting blood glucose levels [4]. Tea is abundant in essential functional components such as tea polyphenols, tea polysaccharides, and tea pigments, which have garnered considerable attention. Extensive studies have consistently demonstrated the remarkable hypoglycemic effects of these components, particularly in mitigating postprandial blood glucose levels and their potential in the prevention and treatment of type 2 diabetes mellitus (T2DM).

Tea, derived from a range of tea varieties, geographical regions, and processing techniques, displays variations in its content, composition, and functional bioactive components. It is worth noting that the categorization of tea into six major classifications based on the degree of fermentation emphasizes these distinctions, resulting in a wide array of hypoglycemic effects and their underlying mechanisms. These variations serve to emphasize that different types of tea possess distinct profiles of bioactive compounds. Their effects on blood glucose regulation can vary significantly, providing an extensive scope for investigating tea’s multifaceted nature and potential health advantages. Presently, most review articles tend to focus on analyzing individual active components in connection with hypoglycemic effects. Nevertheless, there is a shortage of thorough investigations delving into blood glucose regulation from the standpoint of fermentation levels. This is an area where more extensive research and analysis could provide valuable insights into the various health advantages of tea.

Diabetes is a condition that arises from a disturbance in the body’s glucose metabolism. It can adversely affect crucial organs like the heart, blood vessels, and eyes, leading to significant health risks, and in severe instances, it can even become life-threatening [5]. Due to improved living standards and shifts in dietary habits, there has been a notable rise in diabetes cases. The three main types of diabetes are type 1 diabetes mellitus (T1DM), T2DM, and gestational diabetes mellitus (GDM). Of these, T2DM is the most common, representing around 90% of cases. This is a chronic condition that significantly impacts the quality of human life. In 2030, the International Diabetes Federation projects that the global diabetic population will reach an astounding 578 million individuals [6]. Insufficient insulin production by pancreatic β cells or the occurrence of insulin resistance (IR) can lead to the development of T2DM [7]. Lowering postprandial (after-meal) blood glucose levels is crucial in preventing T2DM. Managing diabetes is a lifelong commitment that includes dietary control, regular exercise, and medication to maintain blood glucose levels as close to normal as possible. Furthermore, it has been reported that stimulating the secretion of orexin in the hypothalamus can reduce the body’s metabolism and increase and maintain the body’s energy reserves [8]. In short, lowering postprandial blood glucose levels is beneficial in preventing diabetes and avoiding related complications.

Currently, numerous hypoglycemic medications on the market have potential side effects. Researchers are increasingly focused on finding natural dietary components to prevent diabetes and reduce complications. Tea is a natural, healthful beverage abundant in dietary active ingredients and serves as a significant source for effectively regulating human blood glucose levels. Research on the hypoglycemic effects and mechanisms of tea typically categorizes into several key aspects: First, enhancing insulin secretion and mitigating IR. Second, facilitating glucose absorption in peripheral tissues. Third, decreasing the activity of enzymes related to gluconeogenesis and digestive processes in the body.

In this paper, we comprehensively analyzed various fermented teas to investigate their potential in lowering blood glucose levels. We aimed to better understand how various fermented teas help maintain stable blood glucose levels in humans. The research progress on the effect and mechanism of tea products with different fermentation degrees in regulating T2DM are reviewed. The relationship among different fermentation levels of tea, functional ingredients of tea, and the prevention of T2DM is explored. We hope to provide insights into the optimal utilization of teas with various fermentation levels and their related products for blood glucose reduction, offering consumers informed choices in tea products. These findings advance the study of the hypoglycemic effects of six specific tea types and provide practical guidance for consumers selecting tea products.

To gain a better understanding of the impact of tea on T2DM, we conducted a literature search and collected high-quality articles from the Web of Science Core Collection, PubMed, and Scopus databases in the last twenty years. The search deadline was set for 25 October 2023. Those databases were searched to identify studies containing the keywords ‘tea’, ‘blood sugar’, ‘blood glucose’, and ‘hyperglycemia’.

## 2. The Main Factors of T2DM and Prevention Methods

### 2.1. The Main Factors of T2DM Mellitus

As shown in Figure 1, a diet high in both sugar and fat significantly contributes to the development of T2DM. The primary underlying causes of T2DM are IR and inadequate insulin secretion. IR denotes a diminished metabolic response in target cells, including the liver, muscle, and fat cells. Consequently, this results in the continuous overproduction of insulin by pancreatic β cells. Diminished tissue responsiveness to insulin, coupled with excessive insulin production, results in the development of hyperglycemia and hyperinsulinemia. These factors significantly contribute to the initiation of inflammation associated with T2DM [9,10]. Insulin deficiency denotes an inadequate production of insulin by the pancreas, leading to elevated blood glucose levels. The occurrence of insulin deficiency is closely linked to the apoptosis of pancreatic β cells. The escalation of β cell apoptosis is a pivotal factor in the progression of T2DM. While both IR and insulin deficiency have genetic and environmental factors at play, environmental influences exert a significant influence on the occurrence of IR and insulin deficiency, contributing significantly to their development [10,11].

### 2.2. The Main Prevention and Treatment Methods of T2DM

#### 2.2.1. Exercise Prevention

Exercise includes various forms such as stretching, aerobic activities, and resistance training, and the choice should be determined by factors like age, cognitive function, and physical activity level, especially in diabetic patients. Research has shown that imbalances in post-meal glucose, insulin, and triglyceride levels can lead to oxidative stress, which in turn leads to inflammation. Participating in short activities like walking and basic resistance exercises, rather than prolonged sitting, has shown the potential to reduce the post-meal rise in glucose, insulin, C-peptide (a marker of internal insulin production), and triglyceride levels in T2DM patients, providing relief from oxidative stress and inflammation [12,13]. Interestingly, in a randomized crossover experiment, researchers found that afternoon exercise was more effective in improving blood glucose levels in T2DM compared to morning exercise [14].

In addition, exercise can improve sleep quality and indirectly affect blood sugar level regulation. Healthy sleep is crucial for T2DM management [15]. Maintaining adequate sleep and limiting energy intake can effectively aid weight loss [16]. Another study demonstrated that extending the sleep duration of sleep-deprived individuals enhances insulin sensitivity [17].

#### 2.2.2. Drug Therapy

Drug adjustment therapy is often more effective at reducing blood glucose levels and is categorized into the following groups [18]: (1) Insulin sensitizers, such as biguanides (e.g., metformin) and gliquidones (e.g., thiazolidinediones), enhance glucose uptake in peripheral tissues. (2) Insulin secretagogues, like sulfonylureas (e.g., glimepiride) and glibenclamides (e.g., repaglinide), stimulate insulin secretion by pancreatic β cells. (3) Digestive enzyme inhibitors, with acarbose as a prominent example, reduce glucose absorption in the small intestine by inhibiting the activity of pancreatic α-amylase and intestinal α-glucosidase. This mechanism effectively lowers blood glucose levels. (4) Sodium-glucose cotransporter-2 (SGLT2) inhibitors, such as caragliflozin, work by preventing the reabsorption of glucose in the kidneys, leading to glucose excretion through urine. (5) DPP-4 inhibitors, like sitagliptin, increase the levels of glucagon-like peptide-1 (GLP-1) and glucose-dependent insulinotropic polypeptide (GIP) by inhibiting DPP-4 activity. This, in turn, stimulates insulin secretion by pancreatic β cells. (6) GLP-1 receptor agonists, like liraglutide, activate pancreatic GLP-1 receptors, indirectly promoting insulin secretion. While drug therapy quickly and effectively lowers blood glucose levels with significant results, numerous studies have demonstrated that long-term use of hypoglycemic agents can result in numerous side effects, including flatulence, liver damage, weight imbalance, and hypoglycemia [15,19,20]. As a result, natural plant extracts have shown exceptional efficacy in regulating blood glucose.

#### 2.2.3. Application of Natural Plant Extracts in Blood Glucose Regulation

Natural plant extracts, derived from nature’s abundance, are renowned for their healthful properties. Extensive scientific research consistently validates their potential in mitigating various chronic diseases, especially T2DM resulting from prolonged hyperglycemia. T2DM patients commonly exhibit an increased high-fat dietary intake alongside reduced physical activity, which exacerbates IR [21]. Digestive enzymes like α-amylase and α-glucosidase primarily break down carbohydrates into glucose, which is then absorbed and utilized by the small intestine. During digestion, different carbohydrates like galactose, maltose, and sucrose can stimulate the secretion of GLP-1, helping reduce postprandial blood glucose levels [22]. Moreover, increased protein intake can mitigate inflammation linked to T2DM. For example, glutamine can alleviate inflammation and enhance glucose uptake in skeletal muscle. Also, L-arginine stimulates GLP-1 release and supports in vivo glucose homeostasis [23,24]. Additionally, consuming fatty acids like omega-3, monounsaturated fats, α-linolenic acid, and docosahexaenoic acid has been observed to preserve muscle mass in elderly T2DM patients. Free fatty acids also play a crucial role in regulating GLP-1 secretion [22,25].

The recent literature emphasizes the increasing interest in using natural active ingredients to control postprandial hyperglycemia. These compounds, including polyphenols [26], polysaccharides [7], peptides [27], and dietary fiber [28], have shown significant research progress in managing postprandial hyperglycemia and preventing diabetes and its complications. Being the second most consumed beverage worldwide, tea contains numerous bioactive compounds such as tea polyphenols, tea polysaccharides, tea polypeptides, and tea dietary fibers. Exploring the functional components and activities of tea not only advances the field of tea processing but also aligns with green development principles. This effort has considerable research significance.

## 3. The Biochemical Characteristics and Hypoglycemic Effect of Tea with Different Fermentation Degrees

### 3.1. Biochemical Characteristics of Tea with Different Fermentation Degrees

Green tea, white tea, yellow tea, oolong tea, black tea, and dark tea are the six traditional categories of fermented tea, each with unique characteristics. When producing green tea, the chemical composition closely resembles that of fresh tea leaves because it is not fermented. A critical step in processing green tea is de-enzyming, which is accomplished by applying high-temperature treatment to deactivate polyphenol oxidase. This process stabilizes the chemical components in the tea and imparts the distinct green characteristics commonly associated with green tea [29].

White tea and yellow tea undergo oxidative fermentation levels between those of green tea and oolong tea. White tea production primarily comprises two essential steps: withering and drying. These processes help preserve a substantial amount of white tea polyphenols and amino acids. Particularly, the withering phase is crucial for preserving the aroma in the white tea manufacturing process [30,31]. Moreover, the drying process in white tea production can improve the overall aromatic profile, including ketones and aldehydes levels, while reducing alcohol formation. This process significantly contributes to shaping the aromatic qualities of the tea [31]. Yellowing is the central process in yellow tea production. During the 18 h yellowing stage in yellow tea processing, there is a significant rise in amino acids and sucrose levels. This increase is linked to the breakdown of epigallocatechin gallate (EGCG) and epicatechin gallate (ECG). This phenomenon’s primary mechanism is the oxidation of EGCG’s B-ring phenolic hydroxyl group by polyphenol oxidase, leading to o-quinone formation. In conditions of high heat and humidity, o-quinones tend to undergo further oxidation and polymerization, ultimately resulting in the formation of theaflavins [32,33].

In oolong tea processing, two critical techniques are used: shaking and stir-frying. Shaking involves the gentle collision of tea leaves, causing slight breakage, reducing moisture content, intensifying enzymatic catalysis and hydrolysis reactions, and resulting in the interaction of catechins with specific amino acids, leading to the release of aldehydes or oxidation into polyphenols [34]. Baking is the final stage in oolong tea production, playing a pivotal role in shaping the distinctive flavor profile of this tea variety. Research has shown that it effectively reduces the tea’s bitterness and astringency, enhances its sweet aftertaste, and contributes to the unique flavor that characterizes oolong tea [35].

Black tea, renowned for its full fermentation, is among the most widely consumed tea types. Fermentation is crucial in black tea processing. During fermentation, catechins undergo oxidation in the B-ring, involving coupling, condensation, hydroxylation, decarboxylation, and rearrangement [36]. Theaflavins and thearubigins, oxidation products of tea polyphenols, are vital indicators of the black tea infusion’s color and brightness. These compounds are intricately connected to the actions of polyphenol oxidase and peroxidase. During black tea fermentation, they give rise to the distinct color and flavor defining black tea [36,37].

Post-fermentation is a defining characteristic of dark tea. Pile-fermentation, a crucial step in dark tea production, is pivotal in creating its distinctive color and flavor. During microbial fermentation, catechins, flavonols, and flavonoids are reduced, while the levels of total theabrownins and phenolic acids increase due to enzymatic reactions or microbial metabolism [38]. In this process, pre-treated tea leaves are piled under controlled temperature and humidity, promoting the prolific growth of microorganisms. Enzymes and microorganisms working together facilitate the transformation and decomposition of active substances in the tea [38,39].

The bioactive components in tea, such as tea polyphenols, alkaloids, tea pigments, and amino acids, exhibit significant variation due to differences in processing technology, geographic environments, and the degree of tea leaf fermentation. These components typically constitute 18–36% of tea polyphenols, 3–5% of alkaloids, 0.3–2% of tea pigments, and 2–4% of the tea’s dry weight [40]. As indicated in Table 1, catechins in tea with varying degrees of fermentation predominantly consist of four phenotypic catechins and four non-phenotypic catechins. The phenotypic catechins comprise EGCG, ECG, epigallocatechin (EGC), and epicatechin (EC), while the non-phenotypic catechins encompass gallocatechin gallate (GCG), catechin gallate (CG), gallocatechin (GC), and catechin (C) [41].

Alkaloids like caffeine and theobromine are byproducts of the extensive processing of tea. Their levels fluctuate depending on the fermentation degree, influencing the bitterness and astringency of the tea. Amino acids, such as L-theanine, glutamic acid, and arginine, are the key contributors to the umami taste of tea. Additionally, pigments like theaflavins, thearubigins, and theabrownins give rise to the characteristic color of the infusion in fermented teas like black tea and dark tea.

### 3.2. Effect of Tea Products with Different Fermentation Degrees on Blood Glucose Balance

Figure 2 illustrates that different processing techniques and tea varieties produce varying hypoglycemic effects. Recent research indicates that extracts from black tea have a stronger in vitro inhibitory effect on sucrase-isomaltase when compared to extracts from green tea and oolong tea [43]. Polysaccharides extracted from fully fermented black tea and post-fermented dark tea show superior in vitro antioxidant activity. In particular, black tea polysaccharides are more effective in inhibiting digestive enzymes. Moreover, administering black tea polysaccharides orally has led to a significant improvement in IR in diabetic mice [44]. In a recent study, researchers compared water extracts from stir-fried green tea and congou black tea. The study concluded that green tea was more effective than black tea in regulating lipid metabolism. On the other hand, black tea demonstrated the capacity to improve glucose uptake and metabolism, thereby reducing postprandial blood glucose levels. Furthermore, it had a positive impact on the composition of intestinal microorganisms by increasing the abundance of beneficial bacteria in the body [45]. This study revealed that two Japanese green tea varieties, Indo and Cha Chuukanbohon Nou 6, have potent DPP-4 inhibitory activity. This suggests that they have the potential to reduce the breakdown of insulin hormones by DPP-4 in the intestine, ultimately promoting insulin secretion and potentially aiding in managing elevated postprandial blood glucose levels [46]. Black tea and purple tea have the potential to provide benefits in diabetes management. A study conducted in Brazil discovered that both black tea and purple tea exhibited strong inhibition of α-amylase activity. Furthermore, black tea seemed to be more effective in enhancing glucose tolerance in mice, indicating its potential as a substance for postprandial anti-hyperglycemic use [47].

#### 3.2.1. Study on Hypoglycemic Activity of Tea with Different Fermentation Degrees in Cell Model

The use of cell models like 3T3-L1, Caco-2, C2C12, L6, HepG2, and similar cells is key in studying the impact of teas with different fermentation levels on blood glucose regulation. As shown in Table 2, among these cell lines, 3T3-L1 cells can differentiate into adipocytes, while C2C12 and L6 cells are classified as skeletal muscle cells. Caco-2 cells can differentiate and demonstrate a structure and function similar to that of intestinal epithelial cells, while HepG2 cells are derived from human liver tumors. When studying the blood glucose balance in vivo, Caco-2 and HepG2 cells are regarded as more mature cell models. Caco-2 cells, resembling the microvilli of the small intestine, are commonly used to evaluate the absorption, transport, and metabolism of bioactive substances from differently fermented teas in the intestinal mucosa. Additionally, HepG2 cells, a classical liver metabolism model, are employed to study the effects of different fermented teas on liver metabolism. The purpose of this study is to enhance our understanding of how bioactive components in different fermented teas affect blood glucose balance.

The study aimed to assess the impact of water extracts from various teas, including green, oolong, and black tea, all grown in the same region, on glucose uptake in Caco-2 cells. The results showed that the water extract from green tea had the most pronounced inhibitory effect on glucose uptake in Caco-2 cells. The author attributed this effect to the high catechin content in green tea, including CG and ECG compounds. These catechins were found to decrease the expression of the SGLT1 gene. Therefore, the study proposes the possible use of catechins as additives in food or beverages to mitigate postprandial (after-meal) increases in blood glucose levels [48]. Additionally, their research report emphasized that black tea extract, particularly theaflavins, showed strong inhibition of sucrase-isomaltase activity, with an IC50 of 8.34 μg/mL. Meanwhile, green tea extract had the most significant inhibitory effect on glucose transporters, particularly SGLT1 and GLUT2 [43]. Furthermore, a Korean study also found that green tea extract can reduce the breakdown of sucrose and the absorption of glucose and fructose in Caco-2 cells [49]. Simultaneously, recent studies in Japan have emphasized the inhibitory effects of green tea extracts obtained through three different extraction methods: room temperature water, hot water, and organic reagents. These extracts have shown inhibitory effects on α-glucosidase in both rat and human Caco-2 cells. Specifically, in rat cells, EGCG, a prominent green tea catechin, is the primary contributor to this effect, while in human cells ECG plays a central role. The author contends that ECG demonstrates the most robust inhibition of intestinal glucose uptake and maintains greater stability in the intestinal and bloodstream environments compared to EGCG [50,61,62].

#### 3.2.2. Study on Hypoglycemic Activity of Tea with Different Fermentation Degrees in Animal Models

Furthermore, animal models were also used to study the effects of bioactive compounds from teas with different fermentation levels on in vivo blood glucose regulation. The study primarily focused on four animal models: normal mice, KK-Ay mice, spontaneous diabetic Torri (SDT) rats, and the db/db mouse model. In a brief timeframe, hyperglycemia was induced in normal mice using a combination of a high-sugar and high-fat diet, as well as intraperitoneal injection of streptozotocin, thus establishing an animal model. Persistent hyperglycemia, hyperlipidemia, and hyperinsulinemia are prominent characteristics observed in db/db mice due to their inherent deficiency in leptin receptors [63]. KK-Ay mice are a common model for studying T2DM. These mice exhibit significant symptoms of obesity and diabetes, including elevated levels of random and fasting glucose, urinary protein, and glycated in the bloodstream [64]. SDT rats are a non-obese spontaneous diabetic model that develops insulin-deficient hyperglycemia after about 16 weeks. A high-sugar and high-fat diet can accelerate the formation of hyperglycemia symptoms, making them a spontaneous diabetic model [65,66]. This differs from KK-Ay and db/db mice, which are diabetic models but with a different onset and characteristics. Polyphenols, pigments, and polysaccharides found in green tea and black tea have been shown to have a positive impact on regulating glucose levels. The main methods used in these studies were free drinking and gavage. Table 3 summarizes the effects of various fermented teas and their extracts on blood glucose regulation.

#### 3.2.3. Epidemiological Investigation on Hypoglycemic Effect of Tea with Different Fermentation Degrees

Flourishing economies and advancements in social development have led to increased consumption of high-sugar and high-fat diets, thereby raising the risk of diabetes, as shown in Table 4. A recent 18.5-year prospective cohort study has notably revealed that regularly consuming tea (more than twice daily) or boiled water (more than five times daily) as alternatives to sugar-sweetened or artificially sweetened beverages can effectively reduce mortality rates and lower the incidence of cardiovascular diseases among patients with T2DM [81]. In a case-controlled study, researchers made an intriguing discovery: tea consumption showed a negative correlation with T2DM. The study involved 27,662 participants from eight European countries who, on average, consumed 152 ± 282 g of tea drinks daily. Encouragingly, supplementing tea drinks reduced the prevalence of T2DM by 22%, suggesting that tea consumption is beneficial for preventing T2DM [82]. Additionally, a meta-analysis has shown that interventions involving tea and its extracts can help maintain stable fasting insulin levels in T2DM patients, indicating a potential beneficial role of tea in aiding insulin control for T2DM management [83]. Meanwhile, Japanese researchers conducted a study with 17,413 subjects who daily consumed unfermented green tea, semi-fermented oolong tea, and fully fermented black tea (at least 6 cups per day). Their results revealed a significant 30% reduction in diabetes risk associated with green tea consumption. Conversely, the consumption of oolong tea and black tea showed no significant correlation with diabetes risk [84]. Furthermore, in a randomized, controlled, cross-over study, 16 obese men with IR received glucose followed by 100 mL of black tea. The results showed that black tea consumption reduced peripheral vascular resistance in both upper and lower limbs after glucose intake. This, in turn, improved postprandial blood glucose and insulin concentrations, effectively alleviating IR [85]. Generally, it is important to consider that the degree of tea fermentation, as well as factors like processing methods, dosage control, and the timing of measurements, can produce diverse results. This inconsistency emphasizes the necessity for additional research in this field to reach more conclusive findings.

## 4. The Potential Mechanism of Different Fermented Tea Regulating T2DM

The influence of tea on post-meal blood glucose regulation has been thoroughly investigated, with an emphasis on multiple mechanisms. In particular, the tea’s ability to lower blood glucose levels is demonstrated across varying levels of fermentation. This pertinent mechanism is visualized in Figure 3.

### 4.1. Inhibition of Digestive Enzymes

Carbohydrates are first consumed and enter the stomach and small intestine. Here, α-amylase breaks them down into oligosaccharides, such as maltodextrin. Then, α-glucosidase, secreted by intestinal epithelial cells, further converts these oligosaccharides into glucose. The small intestine absorbs and utilizes this glucose. Inhibiting the activity of α-amylase and α-glucosidase is an effective method for reducing postprandial hyperglycemia, as shown in Figure 3A. Hypoglycemic drugs, such as acarbose and voglibose, are commercially available and designed to selectively inhibit digestive enzymes, effectively lowering postprandial blood glucose levels in diabetic patients and aiding in maintaining glucose balance. A recent discovery by a Japanese researcher suggests that green tea extract can dose-dependently inhibit α-glucosidase secreted by human intestinal Caco-2 cells. Subsequent investigations into individual components showed that ECG (IC50 = 78.4 ± 15.2 µm) had a stronger inhibitory effect compared to EGCG (IC50 = 90.7 ± 15.8 µm). Interestingly, these results differed from those obtained with rat α-glucosidase [50]. Additionally, tea polysaccharides extracted from teas with different levels of fermentation display significant digestive enzyme inhibitory properties. It is important to note that the fermentation level of tea enhances the biological activity of tea polysaccharides. Smaller molecular weight variants show superior hypoglycemic activity [55,56]. Furthermore, fully fermented black tea is notably rich in theaflavins compared to green tea and oolong tea. It demonstrates a stronger inhibitory effect on the hydrolysis of sucrose, maltose, and isomaltose, with IC50 values of 8.34 μg/mL, 16.10 μg/mL, and 21.63 μg/mL, respectively, showing a clear dose-dependent relationship [43]. In general, black tea shows remarkable effectiveness in this context. Moreover, it is feasible to perform a thorough analysis of the structure–activity relationship of particular compounds present in black tea.

### 4.2. Effect on Glucose Transporters

The human glucose transporter family is categorized into two groups: GLUTs in the first category and SGLTs in the second [89]. Among these transporters, GLUT2 plays a crucial role as the primary glucose transporter in both liver cells and the intestines. In the intestines, GLUT2 aids in glucose absorption by translocating to the apical membrane of intestinal cells, helping to lower postprandial blood glucose levels in humans [90]. GLUT4 is predominantly located in human adipocytes and muscle cells. GLUT4 regulates insulin-mediated glucose uptake and primarily resides intracellularly. As postprandial blood glucose levels increase, insulin secretion triggers the relocation of GLUT4 from within the cell to the plasma membrane. This process assists in glucose absorption and the maintenance of the glucose balance [90,91], as shown in Figure 3.

SGLTs consist of two main types: SGLT1, located in the small intestine, aids in glucose absorption, and SGLT2 is mainly found in the kidney, where it plays a crucial role in reabsorbing glucose within renal tubules [92]. SGLT1 mainly facilitates the active transport of glucose and sodium from the intestinal cavity into cells, which then enter the bloodstream. Conversely, the primary role of SGLT2 is to absorb glucose from urine and reintroduce it into the bloodstream to maintain normal blood glucose levels. Inhibiting SGLT2 prevents the kidney from reabsorbing glucose, causing it to be excreted in urine and effectively reducing blood glucose levels. A study by Ni revealed that green tea extract (IC50 = 0.077 mg/mL) had the most potent inhibitory effect on glucose uptake by intestinal Caco-2 cells compared to oolong tea (IC50 = 0.136 mg/mL) and black tea (IC50 = 0.56 mg/mL) extracts. Furthermore, individual catechins suggested that ester catechins played a significant role in this inhibition. Furthermore, following 24 h of treatment, the gene and protein expression levels of the glucose transporter GLUT2 increased, indicating the potential utility of these compounds as dietary supplements to reduce postprandial blood glucose levels [48]. Another report emphasized that green tea extract demonstrated the most substantial inhibition of glucose transporters under both simulated fasting and simulated feeding conditions. The inhibition rates were 12.53% for SGLT1 and 32.62% for GLUT2. Notably, SGLT1 plays a more prominent role in glucose transport during low glucose states, while GLUT2 is dominant in high glucose conditions [43]. Both black tea thearubigins and dark tea theabrownins up-regulate GLUT4 gene and protein expression, effectively reducing postprandial blood glucose levels [57,75]. We can proceed to examine the effects of tea polyphenols, tea polysaccharides, and tea pigments on the inhibition of renal SGLT2. Furthermore, we can delve into the analysis of the structure–activity relationship.

### 4.3. Inhibition of Gluconeogenesis Pathway

The liver, being a primary glucose reservoir in the body, plays a crucial role in regulating blood glucose levels. It primarily regulates glucose levels through two processes: glycogen breakdown (glycogenolysis) and glucose synthesis (gluconeogenesis). Key enzymes for glycogenolysis include GK and GSK, while gluconeogenesis depends on enzymes like PC, PEPCK, FBP, and G6P [93]. As shown in Figure 3C, a rise in post-meal blood glucose prompts the release of insulin by pancreatic β cells, leading to a decrease in postprandial blood glucose levels. Cell and animal studies have shown that tea extracts enhance the expression of genes and proteins related to glycogen synthesis, leading to increased glycogen synthesis, and they also inhibit the activity of enzymes related to gluconeogenesis. For example, Zhou conducted a 5-week intervention using tea extracts on diabetic mice induced by a high-fat diet. The results indicated a positive correlation between the water extracts of stir-fried green tea and congou black tea with GSK expression levels, and a negative correlation with the gluconeogenesis-related enzyme G6P [45]. Another study demonstrated that intervention with white tea water extract in diabetic mice induced by a high-fat diet and streptozotocin activated the AMPK pathway. This activation resulted in the inhibition of G6P expression, effectively suppressing gluconeogenesis and improving IR [94]. Zhao conducted a 4-week intervention using large-leaf yellow tea extract on diabetic mice induced by a high-fat diet and streptozotocin. The intervention notably decreased the levels of thioredoxin-interacting protein (TXNIP), FBP protein, and FBP enzyme activity. This suppression effectively reduced gluconeogenesis and improved the glucose balance [95]. Tea and its extracts have been shown to decrease the expression of PEPCK, G6P, and FBP, thereby inhibiting gluconeogenesis. Additionally, they enhance GSK expression, promoting glycogen synthesis, which ultimately lowers blood glucose levels in both in vivo and in vitro experiments.

### 4.4. Inhibit the Formation of AGEs

AGEs are the outcome of non-enzymatic browning reactions, like protein glycosylation or the Maillard reaction, which occur when carbonyl groups in reducing sugars (such as glucose and fructose) react with free amino groups in proteins, lipids, or nucleic acids. They are regarded as oxidative derivatives linked to diabetic hyperglycemia [96]. Figure 3D illustrates how prolonged postprandial hyperglycemia produces AGEs, which contribute to diabetic hyperglycemia. The Maillard reaction comprises three stages: The first stage involves the reaction between reducing sugars and protein amino acids, resulting in the formation of an unstable initial glycosylation product called a Schiff base. In the second stage, this product undergoes rearrangement or cross-linking to form Amadori products. In the third stage, small carbonyl compounds derived from Amadori products isomerize with arginine and lysine residues of proteins, leading to the irreversible formation of AGEs [97]. In a recent in vitro experiment, it was shown that black tea polysaccharides effectively inhibit the formation of glycosylation products at each stage of non-enzymatic protein glycosylation. The degree of tea fermentation directly affects the molecular weight of tea polysaccharides, which, in turn, influences their biological activity [56]. Black tea, which is fully fermented, undergoes more extensive oxidative fermentation compared to green tea and oolong tea. Nuclear factor erythroid-2-related factor 2 (Nrf2) plays a crucial role as a transcription factor in regulating the cellular defense against oxidative stress and xenobiotics [98]. In a different in vivo experiment, EGCG reduced AGE levels in the plasma and liver of obese mice on a high-fat diet, suppressed AGE receptor expression, and activated the Nrf2 pathway to counteract AGE formation [68]. Catechins and tea polysaccharides play crucial roles in this process, but additional data are needed for further confirmation and a more comprehensive understanding of the structure–activity relationship.

### 4.5. Inhibition of DPP-4 Activity

DPP-4, being a serine protease, cleaves multiple substrates, including incretin, GLP-1, and GIP [99]. Figure 3E illustrates that natural hormone release stimulates pancreatic β cells, leading to insulin secretion and the regulation of postprandial blood glucose levels. Research suggests that tea extract can inhibit DPP-4 enzyme activity, resulting in increased insulin secretion, decreased glucagon secretion, and lower glucose levels. Zhao identified six protein peptides in dark tea extract. Among these, the peptide VVDLVFFAAAK exhibited the highest α-glucosidase inhibitory activity, with an IC50 of 0.04 ± 0.04 mg/mL, surpassing the efficacy of acarbose (IC50 = 1.51 ± 0.23 mg/mL). Additionally, the peptides MSLYPR and QGQELLPSDFK demonstrated significant inhibitory activities against the DPP-4 enzyme, with IC50 values of 1.35 ± 0.15 mg/mL and 3.89 ± 0.22 mg/mL, respectively [100]. Researchers have confirmed that water extracts from green tea exhibit DPP-4 enzyme inhibitory properties in various green tea varieties, primarily due to their high polyphenol content. Further analysis identified epigallocatechin-3-O-(3-O-methyl) gallate, kaempferol-3-O-rutinoside, myricetin-3-O-glucoside/galactoside, and theogallin as newly discovered DPP-4 inhibitors. These natural active compounds can be integrated into food products to mitigate the degradation of incretin hormones, consequently reducing postprandial blood glucose levels [46]. Furthermore, there is potential for research investigating various fermented teas and their components concerning DPP-4 activity. Such research could yield a deeper understanding of the mechanisms underlying their hypoglycemic effects.

### 4.6. Regulation of Gut Microbiota

Intestinal microbial flora are closely linked to human hyperglycemia and play a critical role in maintaining the body’s energy balance. They also play a key role in metabolic conditions such as diabetes. As shown in Figure 3F, a high-sugar and high-fat diet is a significant contributor to T2DM. Recent research showed that a 5-week intervention using water extracts of black tea and green tea in mice on a high-sugar and high-fat diet resulted in a significant increase in beneficial bacteria (*Allobacteria*, *Lactobacillus*, *Turicibacter*). Simultaneously, it led to a reduction in the levels of Clostridium and Bacteroides. Importantly, deeply fermented black tea exhibited a more significant effect in this context [45]. Additionally, administering extracts of both conventional black tea and selenium-enriched black tea to hyperglycemic mice can modify the composition of intestinal microorganisms. This, in turn, enhances the production of SCFAs and aids in alleviating hyperglycemia [69]. Furthermore, when green tea polysaccharides interacted with intestinal flora, they increased the levels of acetic acid, propionic acid, butyric acid, and total SCFAs in the intestinal environment. At the same time, they reduced the biosynthesis of gluconeogenesis-related amino acids, such as leucine, isoleucine, and valine, which in turn lowered endogenous glucose production. Furthermore, green tea polysaccharides increased lysine and tryptophan biosynthesis, promoted insulin secretion, and reduced postprandial blood glucose levels [77]. Polysaccharides derived from teas of varying fermentation levels show significant differences, especially in their molecular weights, monosaccharide composition, and glycosidic bond connections. Wuyi rock tea, categorized as a semi-fermented oolong tea, falls midway between green tea and black tea in terms of its fermentation level. Following a 40-day intervention in high-fat diet-fed mice, polysaccharides extracted from Wuyi rock tea notably elevated the levels of *Lactobacillus* and enriched *Bifidobacterium*, suggesting their ability to enhance glucose metabolism and improve the gut microbiota in vivo [78]. Sitagliptin is a well-established hypoglycemic drug that acts as a dipeptidyl peptide inhibitor. Recent research emphasizes that gut microbiota-produced DPP-4 (mDPP4), mainly originating from Bacteroides, can decrease GLP-1 levels due to intestinal leakage caused by a high-fat diet. Interestingly, sitagliptin exhibited limited effectiveness in inhibiting mDPP-4. Nevertheless, high-throughput screening identified daurisoline-d4 (Dau-d4) as a selective inhibitor of mDPP4. This compound holds the potential to improve glucose tolerance in diabetic mice with intestinal impairments [101]. We can explore how tea extracts with different fermentation levels and their active components impact mDPP4. This research can provide insights into the connection between these components and mDPP4, with the potential to develop a tailored tea product for individuals with high-fat diet-induced hyperglycemia (intestinal damage) to reduce blood glucose levels.

### 4.7. Reduce Oxidative Stress

Numerous studies have shown a robust connection between inflammation caused by oxidative stress and chronic metabolic conditions like T2DM. Oxidative stress represents an imbalance between the production of free radicals and the body’s antioxidant defenses. This imbalance can lower peripheral insulin sensitivity and initiate the development of T2DM [102]. The majority of cells in the human body have an intrinsic defense mechanism primarily dependent on various enzymes, including SOD, CAT, and GSH [103]. Teas of different fermentation levels contain unique bioactive components. In vivo experiments have confirmed that catechins, especially EGCG present in tea, act as a superb source of natural antioxidants. Experimental evidence from animal studies has shown that various fermented teas can effectively alleviate IR induced by oxidative stress, thus promoting and maintaining the glucose balance in the body. Sampath conducted an intervention in mice on a high-fat diet, administering green tea catechin (EGCG = 75 mg/kg). The results showed that, after 17 weeks, there was a significant rise in GSH levels in the liver, kidney, and adipose tissue of the mice. This intervention effectively activated the oxidative stress defense mechanism, maintaining a balanced equilibrium between free radicals and the antioxidant system [68]. Meanwhile, post-fermented dark tea, particularly ripened pu-erh tea, proved effective in alleviating dextran-induced colitis in mice. Following one week of treatment at a dosage of 10 mg/kg body weight per day, there was a decrease in the expression of NADPH oxidase 2 (NOX2) and NADPH oxidase 4 (NOX4), a reduction in myeloperoxidase (MPO) activity, and the restoration of GSH-PX and SOD activity. These positive effects can be primarily attributed to the presence of catechins and polysaccharide components in the tea [104].

## 5. Conclusions and Prospect

Teas of varying fermentation levels display differing biological activity because of their distinct processing methods. Extracts from tea and its derivatives, such as EGCG and tea polysaccharides, efficiently combat hyperglycemia while regulating glucose balance. In these studies, tea extracts were commonly employed in both in vivo and in vitro experiments, closely mirroring real tea consumption. Tea water extract promotes insulin release through the inhibition of digestive enzymes, enhancement of glucose uptake, suppression of gluconeogenesis-related enzymes, inhibition of AGE formation, reduction in DPP-4 enzyme activity, modulation of gut flora, and attenuation of inflammation caused by oxidative stress. This action assists in alleviating IR and reducing post-meal blood glucose levels. Notably, deeply fermented teas, such as black tea, seem to exhibit a more potent hypoglycemic effect. Exploring the potential of black tea fermentation in regulating post-meal blood glucose is warranted. Furthermore, a holistic approach that combines diet, exercise, and medication can assist in preventing or alleviating T2DM. In the future, research should explore the influence of room temperature and cold brewing on post-meal blood glucose. Tea offers health benefits, yet its daily intake among hyperglycemic patients remains controversial. Additional research in both experimental and clinical settings is required to clarify its impact on blood glucose regulation. Additionally, it is essential to investigate the effective and safe dosages of tea, along with its active ingredients of varying fermentation levels, to comprehend its impact on blood glucose regulation in vivo. The dose–effect relationship, structure–activity relationship, and interaction of different functional components in regulating blood glucose are also worthy of further study.

Finally, we should investigate the inclusion of tea derivatives into everyday staples such as tea rice and tea noodles to assess their effects on post-meal blood glucose. Furthermore, it is essential to develop products aimed at lowering post-meal blood glucose levels for hyperglycemic patients. Further exploration of the activities of different fermented teas and mDPP-4 is justified.

## Figures and Tables

**Figure 1 foods-13-00221-f001:**
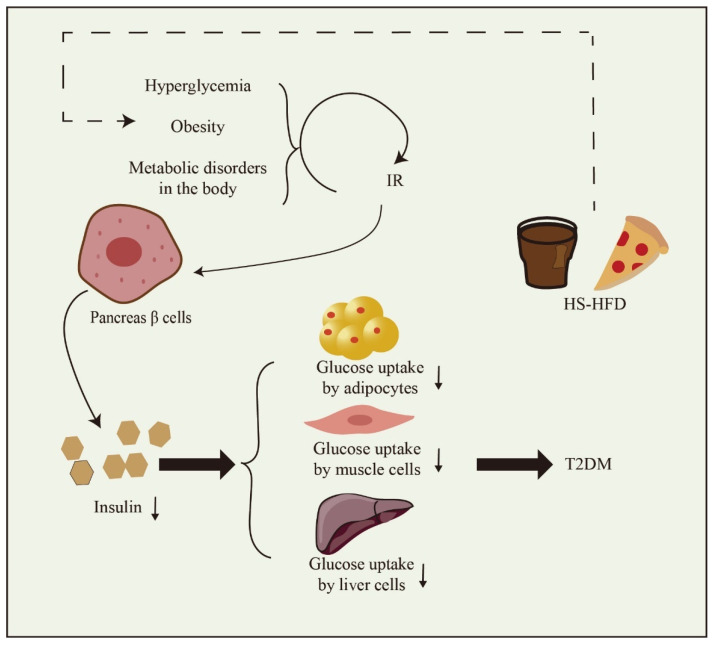
The formation factors of T2DM. High-sugar and high-fat diet (HS-HFD). Downward arrow means down-regulation.

**Figure 2 foods-13-00221-f002:**
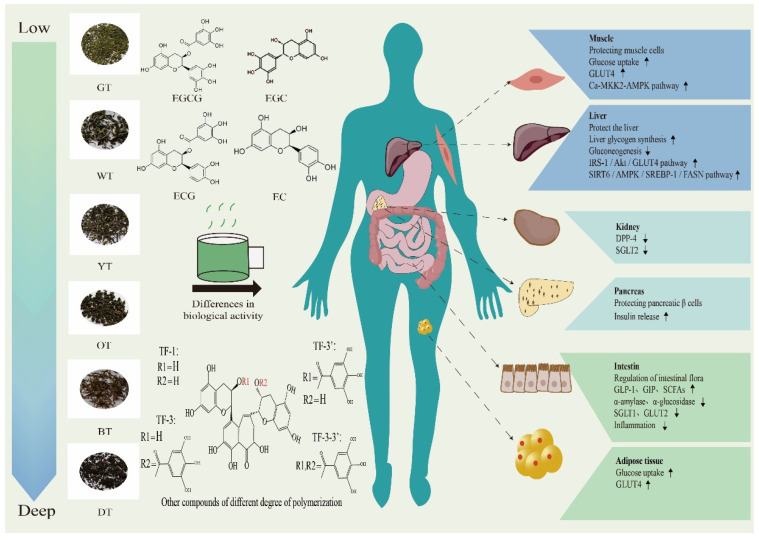
The regulatory effect of different fermented teas on organs of patients with T2DM. Epigallocatechin gallate (EGCG), epigallocatechin (EGC), epicatechin gallate (ECG), epicatechin (EC); theaflavin (TF-1), theaflavin-3-gallate (TF-3), theaflavin-3′-gallate (TF-3′), theaflavin-3-3′-gallate (TF-3-3′); glucose transporters (GLUTs); calmodulin-dependent protein kinase kinase 2 (CaMKK2) -adenosine monophosphate activates protein kinases (AMPK); insulin receptor substrate 1/phosphoinositide 3-kinase/protein kinase B (IRS1/PI3K/AKT); sirtuin 6/adenosine monophosphate activated protein kinase/sterol regulatory element-binding protein-1/fatty acid synthase (SIRT6/AMPK/SREBP-1/FASN); dipeptidyl peptidase-4 (DPP-4); sodium-glucose cotransporter-2 (SGLT2); short-chain fatty acids (SCFAs). Downward arrow means down-regulation. Upward arrow means up-regulation.

**Figure 3 foods-13-00221-f003:**
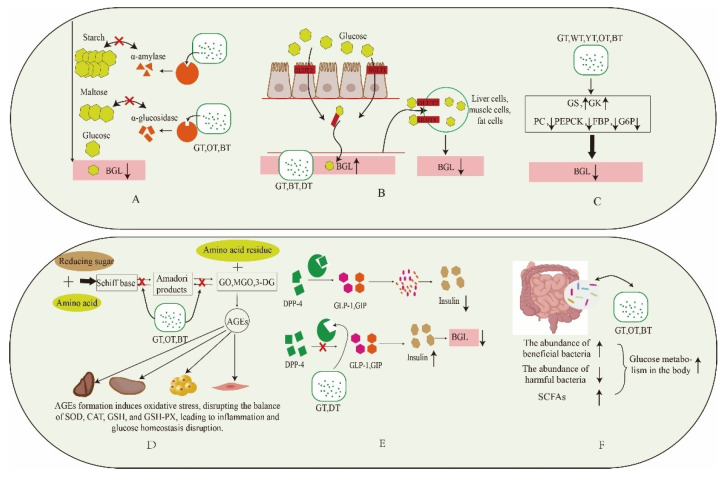
Mechanism of different fermentation degree tea in balancing postprandial blood glucose. (**A**): Inhibition of digestive enzymes. (**B**): Effect on glucose transport. (**C**): Inhibition of gluconeogenesis pathway. (**D**): Inhibition of formation of AGEs. (**E**): Inhibition of DPP-4 activity. (**F**): Regulation of gut microbiota. Blood glucose levels (BGL); glucokinase (GK); glycogen synthase kinase (GSK); pyruvate carboxylase (PC); phosphoenolpyruvate carboxy kinase (PEPCK); gructose-1,6-bisphosphatase (FBP); glucose-6-phosphatase (G6P); glyoxal (GO), methylglyoxal (MGO),3-meoxyglucosone (3-DG); superoxide dismutase (SOD); catalase (CAT); glutathione (GSH); glutathione peroxidase (GSH-PX). Downward arrow means down-regulation. Upward arrow means up-regulation.

**Table 1 foods-13-00221-t001:** The content of bioactive components in tea with different fermentation degrees (mg/g).

	GT	WT	YT	OT	BT	DT	FL	Reference
Total catechins	55.16–198.4	42.81–175.1	52.24–107.9	48.46–159.1	11.39–3.5	30.17–135.9	58.60	[40,42]
Tea polyphenols	238.20	236.36	233.03	224.47	133.46	218.98	245.15	[40,42]
Total free amino acid	21.856	29.642	21.596	26.050	13.382	4.288	28.896	[40,42]
Caffeine	15.6–37.3	15.2–36.8	13.7–38.2	14.4–36.1	13.4–35.7	14.0–36.3	15.5	[40,42]
Theobromine	0.28	0.10	0.17	0.21	0.08	0.19	0.13	[40,42]
Theaflavin	2.34	5.55	2.22	3.68	9.3	2.0	\	[40,42]

Green tea (GT), white tea (WT), yellow tea (YT), oolong tea (OT), black tea (BT), dark tea (DT), and fresh leaves (FL).

**Table 2 foods-13-00221-t002:** Study on hypoglycemic activity of different fermented tea and extracts in cell models.

Tea	Essential Component	Cell Type	Types	Result/Pathway	References
GT, OT, BT	Catechin	Caco-2	Inhibition of glucose uptake	SGLT1 gene expression decreases, and GLUT2 gene and protein expression are inhibited within the first two hours.	[48]
GT, OT, BT, DT	Theaflavins, catechins	Caco-2	Inhibition of digestive enzyme activity and reduction of glucose uptake	BT extract strongly inhibits digestive enzyme activity, while green tea extract has the highest effect on glucose transport (SGLT1 and GLUT2).	[43]
GT	Green tea extract (GTE); green tea polysaccharide (CTP); green tea flavonols(FVN)	Caco-2	Digestive enzyme inhibition; reduce glucose and fructose uptake	GTE can significantly inhibit glucose transport; GTE + CTP + FVN can significantly inhibit fructose transport.	[49]
GT	Catechins	Caco-2	α-glucosidase inhibition	EGCG is the most potent inhibitor for rat α-glucosidase, while ECG is the strongest inhibitor for human Caco-2 cell α-glucosidas.	[50]
Partridge tea	Partridge tea water extract	3T3-L1	α-amylase and α-glucosidase inhibition; enhance glucose uptake	Partridge tea water extract inhibits digestive enzymes, enhances glucose uptake.	[51]
GT	Catechin	3T3-L1	Enhance glucose uptake	GT catechins (EC, EGC) can regulate adipose tissue glucose uptake and lower postprandial blood glucose.	[52]
Mulberry leaf tea extract	Flavonoid	3T3-L1	Enhance glucose uptake	Mulberry leaf flavonoids can reduce the level of free fatty acids and alleviate IR in 3T3-L1 adipocytes./ Activation of IRS1/PI3K/AKT/ GLUT4 pathway.	[53]
BT	Theaflavin	C2C12	Muscle protection	Protect skeletal muscle and maintain muscle health./ Activation of CaMKK2-AMPK signaling axis via Ca^2+^ influx.	[54]
Anji WT	Tea polysaccharide	L6	Inhibit digestive enzyme activity; enhance glucose uptake	Ultrasound-assisted deep eutectic solvent extraction of Anji WT polysaccharide exhibits strong digestive enzyme inhibitory activity, enhancing glucose uptake by L6 cells.	[55]
GT, OT, BT	Tea polysaccharide	L6	α-glucosidase inhibition; inhibition of AGE formation; enhance glucose uptake	The degree of fermentation is directly linked to the biological activity of tea polysaccharides, including antioxidant, anti-glycosylation, α-glucosidase inhibition, and hypoglycemic effects on L6 myotubes.	[56]
BT	Theaflavin	HepG2	Enhance glucose uptake	Theaflavins promote GLUT4 translocation, boost glucose uptake, reduce IR, and enhance mitochondrial biogenesis while reducing adipogenesis./ Activation of IRS-1/Akt/GLUT4 pathway.	[57]
BT	Theaflavin-3,3′-digallate	HepG2 /Zebrafish	Enhance glucose uptake and protect pancreas β cells	Regulation of phosphoenolpyruvate carboxykinase and glucokinase promotes islet β cell regeneration, reducing blood glucose.	[58]
GT, WT, BT	Tea polyphenol	HepG2	Enhance glucose uptake	WT polyphenols had the strongest glucose uptake in HepG2 cells.	[59]
Puer tea	Puer tea water extract	HepG2	Enhance glucose uptake	At a 0.1 g/L concentration, the water extract exhibited higher glucose uptake than 10 μmol/L of acarbose.	[60]

**Table 3 foods-13-00221-t003:** Hypoglycemic effects of different fermented tea and extracts in animal models.

Tea	Main Components	Time, Way	Animal Type	Diabetes Induction Mode	Result/ Pathway	References
Roasted GT, kung fu BT	Tea water extract	5 weeks, free to drink	SPF C57BL/6J male mice	HSFD	It can promote liver glycogen synthesis and inhibit gluconeogenesis./ Activation of IRS-1-PI3K/AKT-GLUT2 pathway.	[45]
GT	Green tea extract	19 days, free to drink	Male Wistar/ST rats; KK-Ay mice	STZ	It can promote GLUT4 translocation in skeletal muscle; enhance glucose uptake.	[67]
GT	EGCG	17 weeks, free to drink	C57BL/6 mice	HFD	It can reduce AGEs to lower plasma glucose and alleviate diabetic complications.	[68]
Selenium-rich BT and ordinary BT	Tea water extract	5 weeks, free to drink	C57BL/6J mice	STZ/HFD	It can inhibit digestive enzymes, regulate glucose metabolism, and alleviate liver injury and inflammation./ Activation of PI3K/Akt pathway.	[69]
Selenium-rich BT, selenium-deficient BT	Tea powder	4 weeks, free intake	Male Sprague-Dawley rats	HFD	It can improve weight, lower total triglycerides and fasting blood glucose, enhance insulin sensitivity, relieve liver and intestinal injury, reduce inflammation, and enrich beneficial intestinal bacteria.	[70]
GT, YT, BT	Water extract	25 days, free to drink	Male ICR mice	HFD	GT and BT maintain body weight in high-fat diet mice, while YT significantly lowers blood glucose.	[71]
BT	Theaflavins, thearubigins	56 days, free to drink	Male Sprague Dawley rats	HSD	Theaflavins and thearubigins reduce plasma glucose and boost insulin release.	[72]
BT, DT	Tea water extract	28 days, gavage (1000, 500, and 300 mg/kg).	Male SPF Kunming mice	Intraperitoneal injection of STZ	Enhances glucose transport, reduces postprandial blood glucose, and inhibits liver glycogen synthesis./ Up-regulation of PI3K, AKT, IRS1, GLUT2; down-regulation of GSK3β protein and gene expression.	[73]
GT, OT, BT, DT	Tea polysaccharide	4 weeks, gavage (0.2 mL, 150 mg/kg, 200 mg/kg, 300 mg/kg)	Male ICR mice	HSFD	BT polysaccharide is highly effective in inhibiting α-glucosidase and reducing triglyceride, total cholesterol, LDL cholesterol, creatinine, alanine aminotransferase, and aspartate aminotransferase.	[44]
GT, WT, OT, BT, purple tea	Tea water extract	Gavage, Samples were taken once every 30 min for two hours (500 mg/kg).	Normal male Swiss mice	Normal male Swiss mice	Purple tea shows the highest starch digestion inhibition. BT shows the most significant improvement in glucose tolerance.	[47]
BT	Theaflavins	7 weeks, oral	Male C5BL mice	HFD	It can reduce blood glucose, improve IR, alleviate liver injury, and lower serum triglycerides, total cholesterol, LDL cholesterol, as well as alanine aminotransferase and aspartate aminotransferase levels./ Activation of SIRT6/AMPK/SREBP-1/FASN pathway.	[74]
Raw Pu‘er tea, cooked Pu‘er tea	Theabrownin	24 weeks, gavage (400 mg/kg)	C57BL/6J male mice	HFD	Both inhibit weight gain and maintain glucose homeostasis./ Up-regulation of GLUT4 and IRS1.	[75]
DT	Tea protein	21 days, gavage (50 mg/kg, 100 mg/kg, 125 mg/kg)	SPF C6BL/8 male mice	alloxan	It can inhibit weight, enhance glucose tolerance, and reduce fasting blood glucose. Activation of spleen–brain axis to alleviate hyperglycemia.	[76]
GT	Tea polysaccharide	3 weeks, gavage (100 mg/kg, 200 mg/kg, 400 mg/kg)	Male Wistar rats	HFD	It can reduce blood glucose, promote SCFAs production, alleviate IR, mitigate pancreatic and liver damage, and increase beneficial intestinal bacteria.	[77]
Wuyi rock tea (OT)	Tea polysaccharide	40 days, gavage	Male Wistar rats	HFD/STZ	It can regulate gut flora, boost beneficial bacteria, lower fasting blood glucose, improve glucose tolerance, and ease liver and pancreatic damage.	[78]
BT	Theaflavins	20 weeks, gavage (25 mg/kg/day)	SDT rats	SDT rats	It can improve glucose tolerance, promote incretin secretion and improve IR	[66]
Tea grounds	Tea dietary fiber	28 days, gavage (250 mg/kg, 500 mg/kg, 1000 mg/kg)	Male wistar rats	HFD/STZ	It can improve hyperglycemia, enhance insulin sensitivity, ease pancreatic injury, boost SCFAs levels, and stimulate insulin secretion.	[79]
L-theanine	L-theanine	28 days, gavage (200 mg/kg)	Male Wistar rats	Nicotinamide/STZ	It can reduce leptin and adiponectin levels in the hippocampus of diabetic mice, alleviate hippocampal damage, promote overall blood glucose metabolism, and maintain the glucose balance in the body.	[80]

High-sugar and high-fat diet (HSFD), streptozotocin (STZ), high-fat diet (HFD), high-sugar diet (HSD), reduced glutathione/oxidized glutathione (GSH/GSSG), glycogen synthase kinase 3β (GSK3β).

**Table 4 foods-13-00221-t004:** The relationship between tea and diabetes in different countries.

Country	Year	Tea, Method	Sample				Results	References
			Total	Males (%)	Females (%)	Average Age		
USA	1980–2018	>2 times/day	15,486	26.4	73.6	61.3	Substituting sugar-sweetened beverages with tea, coffee, or plain water can lower mortality and reduce cardiovascular disease incidence in adults with T2DM.	[81]
Europe	1992–2007	152 ± 282 g/day	12,333	50	50	56.0	Switching to tea from sugary drinks reduces the risk of T2DM by 22%.	[82]
Iran	2009	4 cups/day (600 mL)	23	/	/	57.0	It inhibits serum malondialdehyde, reduces C-reactive protein, increases glutathione levels, and protects the cardiovascular system in diabetic patients.	[86]
Netherlands	2012–2013	100 mL BT	16	100	0	/	Consuming BT in diabetic patients lowers peripheral vascular resistance in limbs after glucose intake, improves postprandial blood glucose, insulin concentration, and mitigates IR.	[85]
Singapore	1999–2004	1 cup/day or more	36,908	/	/	54.8	Daily consumption of BT (>1 cup/day) reduces diabetes risk by 14%.	[87]
Japan	5 years random access	6 cups/day or more	17,413	38.6	61.4	53.0	GT reduces diabetes prevalence by 33%.	[84]
China	1991–2006	5 g/day or more.	164,681	100	/	54.0	GT consumption lowers all-cause mortality and reduces cardiovascular disease risk in healthy adult males compared to non-drinkers.	[88]

## Data Availability

Not applicable.

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
