# Peer review of "Research Progress on the Effect and Mechanism of Tea Products with Different Fermentation Degrees in Regulating Type 2 Diabetes Mellitus"

_foods, 2024, doi:10.3390/foods13020221_

Round 1

Reviewer 1 Report

Comments and Suggestions for Authors

Thank you very much for your interesting research. Some points must be carefully revised:

ABSTRACT. Line 15. '...tea reduces' sounds too categorical. This fact depends on many parameters and conditions. Please, rephrase it.

INTRODUCTION. Tea kombucha increasing consumption and interest worths to be mention in this section, since some studies have demonstrated its hypoglycemic/antidiabetic effect.

CONCLUSION AND PROSPECT. Please, explore and describe the current limitations of the published studies in greater depth.

Reviewer 2 Report

Comments and Suggestions for Authors

In this Review the authors comprehensively analyzed various fermented teas to investigate their potential in lowering blood glucose levels. 

The manuscript is well structured and deals with a topic of great relevance and potential interest for the scientific community. 

However, I have some suggestions for authors.

Major comment

Introduction

Although this paragraph is well structured, I believe that some changes are necessary. In particular, the authors, although the topic covered by the review has been extensively covered, do not provide sufficient bibliographical references. For example, from line 32 to line 56, and from line 65 to line 82, there are no bibliographical references. In this regard, I advise authors to take into consideration the following works:

-       Messina A et al., Role Of The Orexin System On Arousal, Attention, Feeding Behaviour And Sleep Disorders, Acta Medica Mediterranea, 2017, 33: 645;

-       Greenwalt CJ, Kombucha, the fermented tea: microbiology, composition, and claimed health effects. J Food Prot. 2000 Jul;63(7):976-81. doi: 10.4315/0362-028x-63.7.976.

Furthermore, the authors should specify the purpose of the study in detail at the end of the introductory paragraph, and explain why they wanted to carry out a review on this topic.

The authors should include a methods section, in which it should be specified how the bibliographic search was carried out. Specifically, you should declare the keywords used, the scientific databases, the time frame and the inclusion/exclusion criteria used.

The main prevention and treatment methods of T2DM 

Paragraph 2.2.1 is too brief. In consideration of the topic covered, i.e. Exercise prevention, the authors should provide additional information as there are many articles in the literature that deal with the topic.

The same goes for the section “Drug therapy”.

Conclusion

A "Discussions" paragraph should be included in which the authors should summarize the results and what impact they may have. Furthermore, it would be appropriate, again in the discussion section, to specify the limits of the study.

Minor comment

-       On line 41 the acronym T2DM is used directly without first specifying it in this paragraph. The authors specified the acronym in the abstract, however upon first use in the main manuscript it must be specified again.

-       It is necessary to leave a space, after the words, before inserting the reference (example: line 32 ref 1; line 57 ref 2).

-       The authors should improve the captions of the figures and tables. the caption should include a brief description of what is present in the relevant table/figure, and specify the acronyms used within them. Please review all captions.

-       The resolution of Figures 1and 2 is low

-       Check the formatting of the references both in the final paragraph (References) and in the text.

Comments on the Quality of English Language

Reviewer 3 Report

Comments and Suggestions for Authors

I believe that the review article titled "Research progress on the effect and mechanism of tea products with different fermentation degrees in regulating type 2 diabetes mellitus" is suitable with minor corrections.

The biological effects of tea (extracts) in regulating type 2 diabetes and other biological activities are attributed to specific components (chemical compounds). However, the article lacks a compilation of these compounds (specific molecules, not just groups of compounds) and their respective actions. It would be valuable to identify the predominant compounds and their roles in determining specific properties in products with varying degrees of fermentation.

In section 2.2.2, "Drug therapy," there is a lack of literature references.

In section 3.1, "Biochemical characteristics of tea with different fermentation degrees," the chemical transformations (composition of compounds) occurring during the production of different types of tea are insufficiently described.

References 31 and 32 pertain to the entire Table 1, which discusses the content of bioactive components in tea with different fermentation degrees (mg/g). It is unclear whether these references exclusively address caffeine or other components as well.

Overall, I believe the article addresses important issues related to managing one of the prevalent lifestyle diseases.
